# The Effect of Aerobic Fitness on Psychological, Attentional and Physiological Responses during a Tabata High-Intensity Interval Training Session in Healthy Young Women

**DOI:** 10.3390/ijerph20021005

**Published:** 2023-01-05

**Authors:** Cleopatra M. Reppa, Gregory C. Bogdanis, Nektarios A. M. Stavrou, Maria Psychountaki

**Affiliations:** 1School of Physical Education and Sports Science, National and Kapodistrian University of Athens, 41 Ethnikis Antistassis Str., 172 37 Athens, Greece; 2Sport Psychology Lab, Hellenic Sports Research Institute, Olympic Athletic Center of Athens “Spyros Louis”, 151 23 Marousi, Greece

**Keywords:** high-intensity interval training, HIIT, psychophysiological approach, psychological responses

## Abstract

The current study examines the effects of a Tabata high-intensity interval training (HIIT) session on affective, cognitive and physiological indicators in women of different fitness levels. A total of 28 adult women (aged 24.2 ± 1.5 years) completed a 20 m shuttle run test and were then assigned to higher fitness and lower fitness groups (HF and LF, *n* = 14 each) according to their predicted aerobic power. On a separate occasion, participants completed a 30 min Tabata workout (six 4 min rounds separated by 1 min passive rest). Each round included eight exercises (20 s exercise and 10 s rest). Affective, physiological and cognitive responses were assessed prior to, during and after the protocol. Heart rate and blood lactate concentration increased similarly in both groups over time throughout the workout (*p* < 0.001). Total Mood Disturbance was higher for LF (111.4 ± 15.7) vs. HF (102.9 ± 11.7) (*p* = 0.48), vigor showed a level by time interaction of *p* = 0.006 and Activation–Deactivation Adjective Check List factors deteriorated over time (*p* < 0.001). The Concentration Grid Test was better overall for HF (10.5 ± 3.6) vs. LF (8.6 ± 3.6) (*p* = 0.05). The Feeling Scale and Rating of Perceived Exertion worsened similarly in both groups over time (*p* = 0.002 and *p* < 0.001, respectively). Positive and negative affect and arousal did not differ between groups or change over time (*p* > 0.05). These results show that, despite the different levels of aerobic fitness, physiological, metabolic, perceptual and affective responses were similar in the two groups of women during a 30 min Tabata session. This may imply that affective responses during this type of HIIT are independent of aerobic fitness.

## 1. Introduction

High-intensity interval training (HIIT) has received considerable attention in the past few years among athletes and fitness practitioners. HIIT protocols include short bouts of high-intensity exercise alternated with periods of low-intensity exercise or rest and are characterized by a high heart rate and oxygen uptake (typically higher than 80–90% of the maximum) [1]. Due to faster metabolic adaptations and a reduced time commitment, HIIT has become very popular among physically active individuals exercising in the gym in various formats, with Tabata workouts being very popular [2]. Tabata consists of 20 s of high-intensity effort followed by 10 s of passive recovery repeated for eight rounds. This training protocol has appeared to be safe and effective even in sedentary populations, improving blood biomarkers, increasing caloric expenditure and promoting both aerobic and anaerobic fitness [3,4,5]. Nevertheless, conflicting data have supported the notion that Tabata should be mainly addressed in athletes [6].

In addition to the physiological effects of HIIT, there are several studies examining the psychological responses to this type of exercise [7,8,9]. Research has shown that there is a negative relationship between exercise intensity and evoked emotions [10]. Thus, when intensity increases above the ventilatory threshold or critical power, pleasure and affect to exercise are negatively influenced [11,12,13]. To explain this relationship, Ekkekakis [14] proposed the Dual-Mode Theory, which indicates that there is a reduction in enjoyment when people exercise above the respiratory threshold. Increased concentrations of blood lactate combined with increased oxygen consumption are metabolically unsustainable and cause unpleasant feelings in the participants, urging them to stop the exercise [15]. The importance of this theory is that adherence to exercise may be influenced by affective responses, thus rendering them important components of exercise prescription [16,17].

An important determinant of physiological and metabolic responses to an HIIT session may be aerobic fitness [18,19]. Aerobically fit individuals demonstrate greater endurance and a greater ability to repeat high-intensity bouts of exercise with less fatigue compared with those with a lower aerobic fitness level [20,21]. Moreover, HIIT accelerates and enhances the metabolic benefits of exercise, and second, it reduces the time required to commit to training [22]. However, strenuous and excessive exercise may adversely affect cognitive function [23]. Thus, the purpose of the present study was to investigate and compare the physiological, affective and cognitive responses to the commonly used Tabata exercise format in healthy adult women of different fitness levels. It was hypothesized that women with lower aerobic fitness would have higher heart rates, blood lactate levels and ratings of perceived exertion during the same Tabata workout compared with women with higher aerobic fitness, and this would be accompanied with a greater impairment of affective and perceptual responses.

## 2. Methods

### 2.1. Participants

Power analysis (G*Power 3.1.9.7; [24]) indicated that a sample size of 24 individuals was required to obtain an effect size of 0.25 (*α =* 0.05, power = 0.80, correlation among repeated measures = 0.5). For reasons of experimental mortality and further enhancement of the statistical power of the design, 28 women were recruited (Table 1). The participants volunteered to take part in the study and signed an informed consent form after having read the attached information sheet. This study was approved by the Bioethics Committee of the School of Physical Education and Sports Science of the National and Kapodistrian University of Athens (Approval Number: 1254/13-01-2021).

### 2.2. Measurements

Based on the purpose of this study and the research hypotheses, the following instruments were used:

Feeling Scale (FS; [25]). The FS is an 11-point Likert scale bipolar measure of a participant’s affective valence, ranging from “Very Bad” (−5) to “Very Good” (+5), with “Neutral” provided at 0.

The Activation–Deactivation Adjective Check List (ADACL) [26] is a self-report instrument that comprises four subscales: energy (5 items), tiredness (5 items), tension (5 items) and calmness (5 items). Participants answered each item based on a 4-point rating scale (i.e., “vv” to signify “definitely feel”, “v” to signify “feel slightly”, “?” to signify “cannot decide” and “no” to signify “definitely do not feel”). Then, the scale was transformed and scored from “1” representing “definitely do not feel” to “4” representing “definitely feel”. The participants, based on the response scale, were asked to rate the extent to which they experienced each of the affects included in the ADACL instrument. Each ADACL factor’s total score ranged from 5 to 20, and higher values represented higher energy, tiredness, tension and calmness. The ADACL is reliable and a valid instrument to evaluate affective responses in exercise [27]. The ADACL subscales indicated acceptable reliability in the pre- and post-test measure (Cronbach *a* = 0.61 to 0.84).

The Profile of Mood States Short Form (POMS-SF) [28] is a 37-item self-report instrument containing six subscales: tension–anxiety, depression–dejection, anger–hostility, vigor–activity, fatigue–inertia and confusion–bewilderment. Participants responded with a 5-point scale, ranging from 0 representing “not at all” to 4 referring to “extremely”, providing a description of their mood state. Curran [29] and Terry, Lane and Fogarty [30] have supported the internal consistency for the POMS-SF subscales, as well as for the test–retest reliability of the instrument. The POMS subscales indicated acceptable reliability in the pre- and post-test measure (Cronbach *a* = 0.60 to 0.94).

The Concentration Grid Test (CGT) was used to evaluate cognitive performance [31] through a concentration test, which is related to the participants’ affective responses. Moreover, the CGT was used to reveal the effect of exercise intensity on concentration. The CGT revealed a reliability of 0.79.

The Rating of Perceived Exertion (RPE; [32]) is a 15-point single-item scale ranging from 6 to 20, with the anchor score ranging from “No exertion at all” to “Maximal exertion”. A meta-analysis of validity data indicated that the RPE showed acceptable weighted mean validity coefficients with intensity physiological indices [33].

### 2.3. Procedures

Participants visited the laboratory two times, 4–7 days apart. Prior to each visit, they were asked (a) to abstain from vigorous exercise for at least 48 h before the test, (b) to eat at least 3 h before the visit and (c) not to drink caffeinated beverages for at least 8 h before each experimental session. During the first visit, the participants filled in a demographic information form, and they were asked about their previous experience in HIIT programs, and their heights and weights were measured. Afterward, they performed a 20 m shuttle run test in an indoor hall, after test familiarization. Aerobic fitness was estimated from shuttle run performance using a prediction equation of maximum oxygen uptake (VO_2max_) [34].

Participants’ VO_2max_ ranged from 25.5 mL/kg/min to 43.87 mL/kg/min (*M* = 33.39, *SD* = 4.61). Based on the calculated VO_2max_, the participants were separated into two equal groups (*n* = 14) using the median split method. The high fitness group (HF) included participants with a VO_2max_ greater than 32.8 mL/kg/min (*M* = 37.0, *SD* = 3.6 mL/kg/min), whereas the low fitness group (LF) included the ones with VO_2max_ lower than 31.87 mL/kg/min (*M* = 29.8, *SD* = 1.8 mL/kg/min). According to the percentile norms set by ACSM [35], which considers gender, average age (*M* = 24.4, *SD* = 1.1) and the VO_2max_, HF was classified in the 50th percentile and LF (*M* = 24.0, *SD* = 1.9) was classified in the 10th percentile of the population.

The main experimental condition occurred 4–7 days after the aerobic fitness test (Figure 1). The volunteers were prompted to perform the aerobic test between the 5th and 10th day of their menstrual cycle, whereas the main experimental trial was performed between the 9th and 14th day (follicular phase). A recent study [36] showed that menstrual cycle does not affect women’s physiological performance, but their psychological responses are weakened in the luteal phase, especially during high-intensity exercise.

Initially, each participant completed the following questionnaires in the pre-test measure: Profile of Mood States (POMS), Positive and Negative Affect Schedule (PANAS), Activation–Deactivation Adjective Check List (ADACL), Feeling Scale (FS), Felt Arousal Scale (FAS), Rating of Perceived Exertion (RPE) and Concentration Grid Test (CGT). Afterward, a baseline value of blood lactate concentration was obtained. The researchers then placed a Polar H10 (Polar Electro Oy, Kempele, Finland) on each participant’s torso, and heart rates started being measured. Immediately after, the experimental protocol was initiated, starting with 5–7 min of aerobic warm-up, which included mobility exercises and dynamic–ballistic stretching of the respective joints and muscle groups. A certified instructor executed the warm-up routine and guided the participants so that they could execute the movements correctly and to keep the intensity at around 70% of their peak heart rate, as recommended before exercising with a Tabata protocol [37]. This was followed by the main workout, which consisted of high-intensity exercises executed for 20 s, interspersed with 10 s of passive recovery (no physical activity). All exercises used body weight as resistance and activated large muscle groups. Six consecutive rounds (4 min each) with a 1 min passive rehabilitation break in between were performed. The exercises were performed in the following order: jumping jacks, mountain climbers, squat jumps, plank jacks, high knees, low skipping with alternating front punches, seal jacks and cross mountain climbers. The researchers demonstrated and explained the exercises to the participants, who were allowed some time after the warm-up to execute them and familiarize themselves with them on a technical level. Music at a rhythm of 96 beats per minute (bpm) was played from a sound system at 57 Watts (47.5 dBm) with songs especially modified and adapted to match the tempo of the Tabata protocol. This circumstance served the participants in order to maintain the same tempo during the execution of each exercise, regardless of their individual abilities. This circumstance was monitored by an experienced instructor, who coached the volunteers throughout the protocol. After each round, participants rested passively for 1 min and responded in one word to the FS, FAS and RPE [38]. A 2 min break was inserted between the 3rd and 4th round in order to measure the blood lactate concentration of the participants. Upon completion of the training protocol and after the single-item questionnaires were directly answered again (FS, FAS and RPE), each participant again completed the following questionnaires approximately 1 min after the completion of the training protocol (post-test): POMS, PANAS, ADACL, FS, FAS, RPE and CGT [8]. After the post-test assessments, participants completed a 5 min cool-down period followed by static stretching for all major muscle groups. About 5 min after recovery, participants completed the CGT questionnaire.

### 2.4. Statistical Analysis

The statistical analysis was performed using IBM SPSS ver.27 Statistics software. The following analyses were implemented: (i) a two-way ANOVA (2 Fitness levels X 2 Time points) for the PANAS, POMS and ADACL factors, (ii) a two-way ANOVA (2 Fitness levels X 8 Time points) for the data of FS, FAS and RPE, (iii) a two-way ANOVA (2 Fitness levels X 3 Time points) for the data of CGT and blood lactate concentrations and (iv) a two-way ANOVA (2 Fitness levels X 6 Time points) for the mean heart rate values of each round in the Tabata protocol. Because the two experimental groups had an equal number of individuals, the condition of homogeneity was ignored [39]. When the assumption of sphericity was violated (Mauchly’s test), the degrees of freedom were adjusted using the conservative Greenhouse–Geisser correction, and the adjusted degrees of freedom were presented for FS, FAS, RPE and HR. When the results of the ANOVAs showed a statistically significant interaction or main effect, the means were compared using Bonferroni post hoc tests. The statistical significance of the examined differences and correlations was set to *p* < 0.05.

## 3. Results

### 3.1. Heart Rate (HR)

The mean HR for each of the six rounds of the Tabata protocol was calculated and expressed as a percentage of peak HR. The two-way ANOVA showed that the main effect of time was significant [*F*
_(1, 26)_ = 15.333, *p* < 0.001, *η*^2^_p_ = 0.38], yet there was no time x fitness level interaction [*F*
_(1, 26)_ = 1.687, *p* = 0.182, *η*^2^_p_ = 0.29] nor a main effect of the level [*F*
_(1, 26)_ = 0.557, *p* = 0.557, *η*^2^_p_ = 0.02]. Pairwise comparisons indicated that mean HR increased in each round and peaked during the 5^th^ round (89.6% ± 5.7% HRpeak, Figure 2).

### 3.2. Blood Lactate Concentration

The analysis of variance showed a statistically significant main effect of time [*F*
_(1, 26)_ = 195.859, *p* < 0.001, *η*^2^_p_ = 0.88]. Pairwise comparisons revealed a significant difference between the pre-test and the mid-test (*p* < 0.001) as well as between the pre-test and the post-test measures (*p* < 0.001). Blood lactate concentration peaked at the mid-test and remained at the same level at the post-test measure (Figure 3). No significant time X fitness level interaction [*F*
_(1, 26)_ = 0.768, *p* = 0.475, *η*^2^_p_ = 0.06] and fitness level main effects [*F*
_(1, 26)_ = 0.371, *p* = 0.548, *η*^2^_p_ = 0.01] were revealed.

### 3.3. Rating of Perceived Exertion (RPE)

The statistical analysis showed only a significant main effect of time [*F*
_(2.616, 68.016)_ = 31.627, *p* < 0.001, *η*^2^_p_ = 0.55]. Pairwise comparisons showed a significant increase in RPE throughout exercise, with the highest value (7.7 ± 1.9) reported at the end of exercise (Figure 4). However, no significant time X fitness level interaction [*F*
_(1, 26)_ = 0.920, *p* = 0.425, *η*^2^_p_ = 0.03] and fitness level main effects [*F*
_(1, 26)_ = 0.256, *p* = 0.617, *η*^2^_p_ = 0.01] were revealed.

### 3.4. Feeling Scale (FS)

The analysis of variance showed only a significant main effect of time [*F* _(7, 20)_ = 6.127, *p* = 0.002, *η*^2^_p_ = 0.19]. Pairwise comparisons showed a significant difference between the values obtained at rest 4 vs. 5 *p* = 0.034. There was also a significant difference between the values derived from rest 6 vs. the post-test (*p* < 0.001). The most important difference, however, was spotted between the pre-test and the value obtained immediately after the last round (*M* = 2.07, *SD* = 1.7 vs. *M* = 0.43, *SD* = 3.1, *p* = 0.018), showing a decline of affect after the last round (Figure 5 ). The main effect of the level revealed a trend of significance [*F*
_(1, 26)_ = 3.064, *p* = 0.092, *η*^2^_p_ = 0.11], and there was no time X fitness level interaction (*F* _(5, 22)_ = 1.577, *p* = 0.210, *η*^2^_p_ = 0.06).

### 3.5. Profile of Mood States (POMS)

The two-way ANOVA for the Total Mood Disturbance (TMD) showed a statistically significant level main effect [*F*
_(1, 26)_ = 4.320, *p* = 0.048, *η*^2^_p_ = 0.14], and LF scored higher than HF during the experimental protocol (Table 2). However, no significant time X fitness level interaction [*F*
_(1, 26)_ = 1.107, *p* = 0.302, *η*^2^_p_ = 0.04] nor time effect [*F*
_(1, 26)_ = 0.203, *p* = 0.656, *η*^2^_p_ = 0.01] were revealed.

The analysis of variance for depression subscale showed a statistically significant main effect of time [*F*
_(1, 26)_ = 9.176, *p* = 0.005, *η*^2^_p_ = 0.26], and the values decreased for both groups in the post-test measure. Moreover, the main effect of level [*F*
_(1, 26)_ = 5.178, *p* = 0.031, *η*^2^_p_ = 0.17] was statistically significant, with LF indicating higher values than those of HL (Table 2). However, there was no time (pre-test and post-test) X fitness level interaction [*F*
_(1, 26)_ = 0.286, *p* = 0.597, *η*^2^_p_ = 0.01]. For confusion, the results showed a statistically significant main effect of time [*F*
_(1, 26)_ = 5.534, *p* = 0.026, *η*^2^_p_ = 0.18], with the pre-test confusion values being higher for both groups than those of the post-test ones (Table 2). However, no significant time X fitness level interaction [*F*
_(1, 26)_ = 0.221, *p* = 0.642, *η*^2^_p_ = 0.01] and fitness level [*F*
_(1, 26)_ = 0.046, *p* = 0.832, *η*^2^_p_ = 0.00] were revealed in the confusion factor. Regarding fatigue, the results showed a significant main effect of time [*F*
_(1, 26)_ = 36.262, *p* < 0.001, *η*^2^_p_ = 0.58], with the post-test values being significantly increased compared to the pre-test values (Table 2) [time X fitness level: *F*
_(1, 26)_ = 1.672, *p* = 0.207, *η*^2^_p_ = 0.06; fitness level: *F*
_(1, 26)_ = 2.969, *p* = 0.097, *η*^2^_p_ = 0.10]. In the vigor subscale, the findings showed a significant time X fitness level interaction [*F* _(1, 26)_ = 9.039, *p* = 0.006, *r* = 0.51, *η*^2^_p_ = 0.26], suggesting that vigor was different in HF and LF. Post hoc tests were performed to compare each fitness level after the experimental protocol, and these tests showed a significant difference when comparing LF to HF [*F*
_(1, 26)_ = 6.342, *p* = 0.018, *η*^2^_p_ = 0.20]. Moreover, a significant difference was revealed in the pairwise comparison upon examining the measurements in each time point for HF [*F* _(1, 26)_ = 7.299, *p* = 0.012, *η*^2^_p_ = 0.22] (Table 2) [time: *F*
_(1, 26)_ = 0.663, *p* = 0.423, *η*^2^_p_ = 0.03; fitness level: *F*
_(1, 26)_ = 1.777, *p* = 0.194, *η*^2^_p_ = 0.06]. The analysis for anger and tension did not show any statistically significant time x fitness level interactions or main effects (*p* > 0.05) [anger: time X fitness level: *F*
_(1, 26)_ = 0.011, *p* = 0.916, *η*^2^_p_ = 0.00; time: *F*
_(1, 26)_ = 0.674, *p* = 0.01, *η*^2^_p_ = 0.10; fitness level: *F*
_(1, 26)_ = 0.195, *p* = 0.662, *η*^2^_p_ = 0.01; tension: time X fitness level: *F*
_(1, 26)_ = 0.278, *p* = 0.603, *η*^2^_p_ = 0.01; time: *F*
_(1, 26)_ = 0.078, *p* = 0.782, *η*^2^_p_ = 0.00; fitness level: *F*
_(1, 26)_ = 0.958, *p* = 0.337, *η*^2^_p_ = 0.04].

### 3.6. Activation–Deactivation Adjective Check List (ADACL)

The statistical analysis showed a significant main effect of time for tension [*F*
_(1, 26)_ = 90.374, *p* < 0.001, *η*^2^_p_ = 0.78], with the values decreasing for both groups in the post-test compared to the pre-test values (Table 2). However, no statistically significant time X fitness level interaction [*F*
_(1, 26)_ = 0.085, *p* = 0.773, *η*^2^_p_ = 0.00] nor fitness level main effect [*F*
_(1, 26)_ = 1.804, *p* = 0.191, *η*^2^_p_ = 0.07] were revealed. Regarding the calmness factors, the two-way ANOVA 2 × 2 showed only a significant main effect of time [*F*
_(1, 26)_ = 32.494, *p* < 0.001, *η*^2^_p_ = 0.56], with both groups presenting higher values before the experimental protocol started than those at the end (Table 2). However, no statistically significant interaction was revealed [time X fitness level: [*F*
_(1, 26)_ = 0.192, *p* = 0.665, *η*^2^_p_ = 0.01], whereas fitness level [*F*
_(1, 26)_ = 1.902, *p* = 0.180, *η*^2^_p_ = 0.07] approached statistical significance. Additionally, the analysis performed for tiredness showed statistical significance for the main effect of time [*F*
_(1, 26)_ = 14.513, *p* < 0.001, *η*^2^_p_ = 0.36], revealing that the values were lower in the post-test measures compared to those of the pre-test (Table 2) [time X fitness level interaction [*F*
_(1, 26)_ = 0.010, *p* = 0.921, *η*^2^_p_ = 0.00; fitness level: *F*
_(1, 26)_ = 3.754, *p* = 0.064, *η*^2^_p_ = 0.13]. Finally, the ANOVA for energy showed no statistically significant time (pre-test and post-test) X fitness level interaction *F*
_(1, 26)_ = 1.501, *p* = 0.232, *η*^2^_p_ = 0.06] nor main effect for fitness level [*F*
_(1, 26)_ = 1.040, *p* = 0.317, *η*^2^_p_ = 0.04], even though there was a trend for a statistically significant main effect of time [*F*
_(1, 26)_ = 3.322, *p* = 0.080, *η*^2^_p_ = 0.11].

### 3.7. Concentration Grid Test (CGT)

The ANOVA results showed a significant main effect of the level [*F*
_(1, 26)_ = 4.234, *p* = 0.050, *r* = 0.37, *η*^2^_p_ = 0.14], with HF showing an overall higher score in the concentration test (*M* = 10.5, *SD* = 3.6) compared to that of LF (*M* = 8.6, *SD* = 3.6) (Figure 6).

### 3.8. Correlation among Physiological Indices, Attention and Affective Responses

The relationship among physiological indices (HRpeak and blood lactate concentration) and attention (GRD) with participants’ affective responses were also examined. The results did not show any significant correlation, except between RPE and fatigue (*r* = 0.53, *p* < 0.01) and between RPE and vigor (*r* = −0.46, *p* < 0.05). In the post-test measure, blood lactate concentration showed a positive correlation with calmness (*r* = 0.42, *p* < 0.05) and a negative correlation with tension (*r* = −0.42, *p* < 0.05). CGT showed a negative correlation to fatigue (*r* = −0.39, *p* < 0.05). Finally, RPE showed negative correlations to vigor (POMS) and energy (ADACL) (*r* = −0.51, *p* < 0.01, *r* = −0.62, *p* < 0.001) and positive correlations to anger (*r* = 0.44, *p* < 0.05), fatigue (*r* = 0.67, *p* < 0.001), confusion (*r* = 0.47, *p* < 0.05) and TMD (*r* = 0.72, *p* < 0.001).

## 4. Discussion

The aim of the present study was to evaluate the effects of the HIIT protocol (Tabata) on the affective, cognitive and physiological parameters of healthy women of different fitness levels. Sex-specific effects of training performance during HIIT protocols have not been adequately investigated, despite the anthropometric and physiological differences between males and females. Moreover, some studies have investigated female samples without controlling for the menstrual cycle, thus impairing the generalization of data for women [36,40,41]. Although any woman could voluntarily participate in the study, the sample was not totally randomized. In that way, the external validity of our study is limited. Therefore, future designs are encouraged to randomly recruit both men and women and match their physical fitness parameters. The main finding of this study was that, despite the different aerobic fitness levels of the two groups, physiological and psychological responses were similar. The Dual-Mode Theory [14] proposes that the relative intensity of exercise is the most important parameter in making a training session pleasant (when the intensity is lower than the respiratory threshold) or unpleasant (when the intensity is greater than the respiratory threshold). Our data show that relative exercise intensity, expressed as a percentage of HRpeak, was high (i.e., between 80 and 90% HRpeak) and similar in the two groups (Figure 2). Furthermore, blood lactate concentration was increased to around 12 mM after the third round of the protocol and remained at that level until the end of exercise in both groups (Figure 3). Similar HR and blood lactate levels in the two groups, irrespective of different aerobic fitness, have also been recently reported during HIIT, where no correlation was found between VO_2max_ levels and blood lactate concentration [15]. Therefore, the similar psychological responses of the two groups may be explained by the fact that exercise stress during this Tabata session was irrespective of aerobic fitness, as assessed by the shuttle run test. It is possible that, despite the same tempo of execution of the exercises, the intensity of each movement was modified according to the fitness level of each participant, thus achieving the same level of effort, as indicated by the identical RPE (Figure 4).

Our data are in line with previous research employing such protocols, reporting that, in females, the increase in RPE during HIIT is rapid, especially when compared to moderate-intensity continuous training (MICT) protocols [22]. RPE after round 3 was between 6 and 8, which translate to “hard” to “very hard” exertion [32], characterizing the “Tabata” sessions [2]. Parallel to RPE, FS ratings decreased over time during exercise. This finding is expected because RPE and FS ratings are inversely related [42]. The deterioration of affect during the HIIT protocol is a common finding [12], and improvements in affect post-exercise may be explained by the fact that, after the completion of exercise, there are feelings of accomplishment [43], and hormonal changes, such as increased endorphins, may also contribute to improved affect post-exercise [44]. The trend of differences in affective responses between the two groups (Figure 5 may indicate a possible differentiation of affect based on fitness level, as reported in the past for individuals with higher physical fitness, who experienced greater pleasure after intense exercise protocols [18].

Mood and affective responses significantly deteriorated after exercise for all participants. Mood factors, such as confusion, depression and fatigue, demonstrated a robust decline post-test (*p* = 0.011 ± 0.013), and this is in line with the existing literature stating that HIIT deteriorates negative mood factors [44]. Interestingly, significant differences in Total Mood Disturbance and depression were found between the two groups, with LF showing higher values (*p* = 0.026 ± 0.030). In combination with the low arousal rates reported by the FAS and the decline in energy, tension and calmness showed by ADACL, it is assumed that this group experienced escalating feelings of sadness and boredom throughout the experimental protocol [45]. Moreover, the only mood factor showing a significant interaction between fitness levels and assessed time points during HIIT was vigor, demonstrating that there is a difference in the way participants experience positive emotions and that they depend on fitness level. This fact contradicts the general perception that HIIT negatively affects the moods of its participants [46].

One interesting finding of this study was the significant difference in concentration between the two groups. HF had better results both post-test and in the follow-up measurement 10 min later. It is also noted that HF had a higher score even in the pre-test, and an explanation for this difference perhaps lies in the fact that HF’s better physical fitness may have contributed to better cognitive functioning, because an increased concentration is a benefit of good physical fitness [47,48]. Contrary to the previous literature supporting that strenuous and excessive exercise may adversely affect cognitive function [23], our results show no statistical difference between time points (Figure 6). The maintained cognitive function during and after exercise may suggest that fatigue caused by this HIIT protocol does not interact with cognitive performance, because HF does not improve their cognitive function, as recent research suggested [49].

## 5. Conclusions

Aerobic fitness level does not seem to influence physiological and psychological parameters during a Tabata HIIT protocol in healthy women. Similar physiological responses (HR and blood lactate levels) indicate that individuals exerted different levels of effort during such protocols, despite the same tempo of exercise execution. Due to the intense nature of this type of exercise, RPE increased up to the “very hard” level, and affective responses deteriorated as exercise progressed. Individuals with higher aerobic fitness seemed to have higher cognitive performance, but this was not affected by exercise. The most important finding of this study, which promotes the knowledge concerning HIIT, is that the sample was split into two experimental groups based on their level of physical fitness, and the two groups showed differences in VO2max and possibly in the associated physiological parameters of respiratory and lactate thresholds. Their psychological responses did not show robust differences as expected. Future studies should examine the psychological effects of a long-term (over eight weeks) HIIT protocol executed by both men and women of different physical fitness levels, which should be assessed using more indices, such as respiratory or lactate thresholds.

## Figures and Tables

**Figure 1 ijerph-20-01005-f001:**
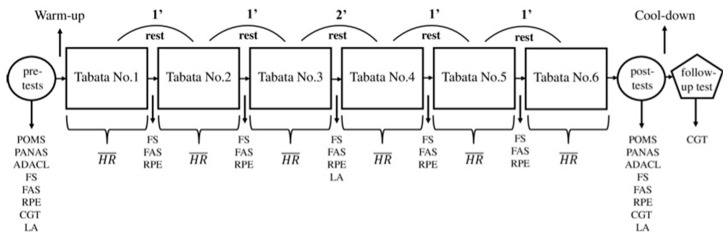
Schematic representation of the main experimental protocol. POMS: Profile of Mood States, PANAS: Positive and Negative Affect Schedule, ADACL: Activation–Deactivation Adjective Check List, FS: Feeling Scale, FAS: Felt Arousal Scale, RPE: Rating of Perceived Exertion, CGT: Concentration Grid Test, LA: Blood lactate concentration, $\bar{\mathrm{HR}}$: mean heart rate.

**Figure 2 ijerph-20-01005-f002:**
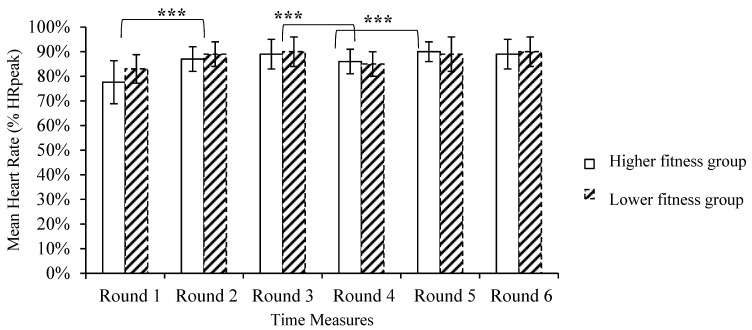
Mean heart rate (HR), standard deviations, and changes expressed as a percentage of peak HR (HRpeak) for the HF and LF groups in each of the six rounds of the protocol (*** *p* < 0.001 between Tabata rounds).

**Figure 3 ijerph-20-01005-f003:**
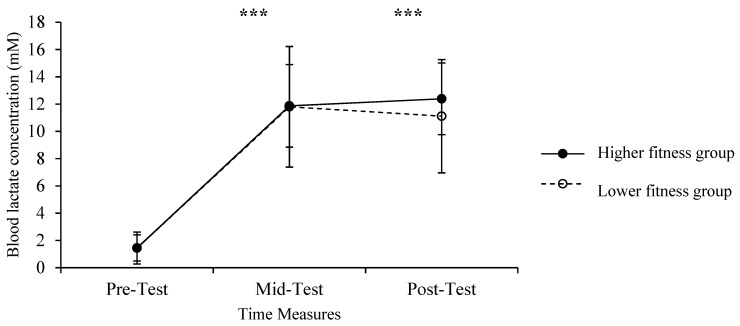
Means, standard deviations and changes in blood lactate concentration in response to HIIT protocol for the high fitness and low fitness groups in pre-test, mid-test and post-test measures (*** *p* < 0.001).

**Figure 4 ijerph-20-01005-f004:**
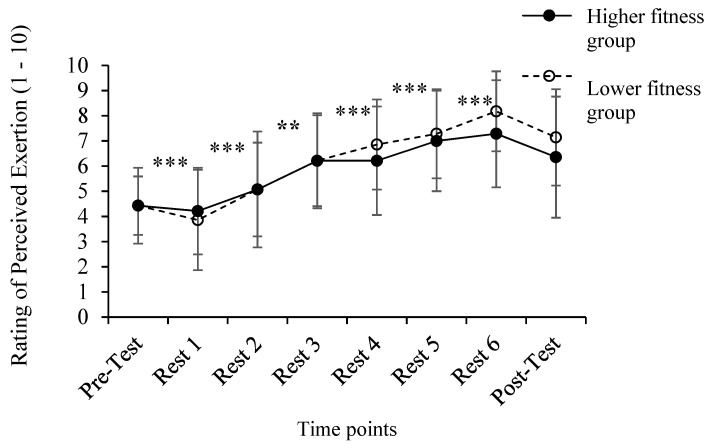
Means, standard deviations and changes in perceived exertion in response to HIIT protocol for high fitness and the low fitness groups before and after the Tabata protocol and during the resting periods separating each round (rest 1 to 6). (** *p* < 0.01, *** *p* < 0.001 between assessed time points).

**Figure 5 ijerph-20-01005-f005:**
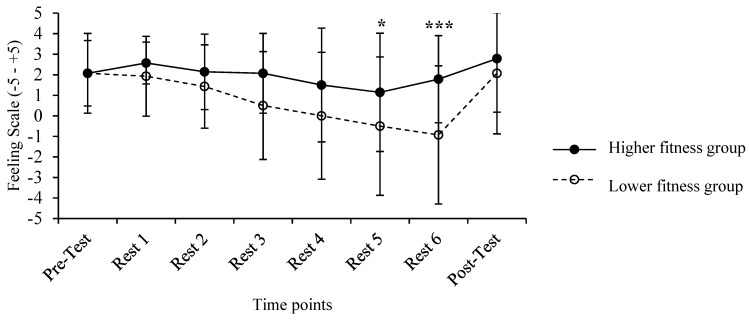
Means, standard deviations and changes in affective valence in response to HIIT for the high fitness and low fitness groups before and after the Tabata protocol and during the resting periods separating each round (rest 1 to 6). (* *p* < 0.05 between rest 4 and rest 5; *** *p* < 0.001 between pre–test and rest 6).

**Figure 6 ijerph-20-01005-f006:**
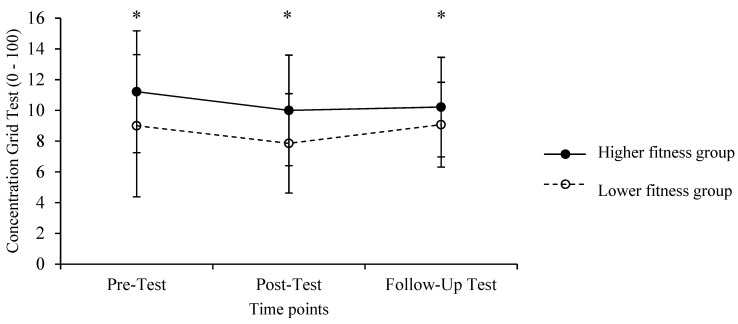
Means, standard deviations and changes in concentration in response to HIIT protocol for the high fitness and the low fitness groups in pre–test, mid–test and post–test measures (* *p * < 0.05 between levels).

**Table 1 ijerph-20-01005-t001:** Demographic and anthropometric characteristics of the participants (*N = 28*).

Characteristics	*M ± SD*
Age	24.2 ± 1.5
Participation in exercise programs (years)	10.3 ± 5.6
Experience in HIIT programs (years)	1.1 ± 1.3
Height	1.63 ± 0.05
Weight	56.8 ± 6.8
BMI	21.3 ± 2.2

**Table 2 ijerph-20-01005-t002:** Means (*M*) and standard deviations (*SD*) of pre-test and post-test for the Profile of Mood States (POMS) and the Activation–Deactivation Adjective Check List (ADACL) for the higher (HF) and lower (LF) fitness groups.

Dependent Variable	HF		LF	
	Pre-Test	Post-Test	Pre-Test	Post-Test
	*Μ* ± *SD*	*Μ* ± *SD*	*Μ* ± *SD*	*Μ* ± *SD*
Total Mood Disturbance (POMS)	103.9 ± 12.2	101.9 ± 11.5	108.9 ± 15.8	113.9 ± 15.7
Depression (POMS)	1.6 ± 2.1	0.1 ± 0.4	3.9 ± 4.5	1.7 ± 2.3
Confusion (POMS)	4.2 ± 3.0	2.9 ± 2.2	4.7 ± 3.6	2.8 ± 2.2
Fatigue (POMS)	3.9 ± 2.6	8.5 ± 5.0	4.8 ± 3.7	11.9 ± 4.7
Vigor (POMS)	12.8 ± 3.5	17.1 ± 4.0	13.8 ± 6.4	11.3 ± 7.7
Tension (ADACL)	16.7 ± 2.6	9.5 ± 2.0	15.4 ± 3.9	8.6 ± 2.6
Calmness (ADACL)	13.7 ± 3.0	9.0 ± 3.2	15.3 ± 3.5	9.8 ± 3.3
Tiredness (ADACL)	14.9 ± 1.8	12.1 ± 1.4	14.2 ± 2.3	11.6 ± 2.5

## Data Availability

The data presented in this study are available on request from the 1st author. The data are not publicly available due to privacy and ethical restrictions.

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
