# Peer review of "The Effect of Aerobic Fitness on Psychological, Attentional and Physiological Responses during a Tabata High-Intensity Interval Training Session in Healthy Young Women"

_ijerph, 2023, doi:10.3390/ijerph20021005_

Round 1
Reviewer 1 Report
The Article “The Effect of Aerobic Fitness on Psychological, Cognitive and Physiological Responses during a Tabata High Intensity Interval Training Session in Healthy Young Women ” is interesting and innoviate but, requires several changes to improve. Authors should pay more attention on details to increase the rigorism of the article. There are some main concerns that need to be clarified.
Major concerns:
Introductions:
1. Authors need to add more details and references in this part, such as the literatures about the effects of acute HIIT or acute Tabata HIIT on affective, cognitive and physiological indicators. Besides, the research gap needs to add in the last paragraph, before the purpose of this study.
2. Why did the authors choose “healthy young women” as the participant? Please explain and add supported references.
Methods:
3. Why did the experiment design lack a control group?
4. Why the authors used a lot of affect-related questionnaires? Which questionnaire could reflect the affect most?
5. Why the authors examined the visual focus? Does this have an association with affect? Authors should add reasons in the introduction part or in the methods part. Besides, could CGT task represent cognitive function? Maybe regard this as attention is more appropriate. Authors could also consider revising the title.
Results:
6. I would suggest authors examine the correlations among affect, attention, and physiological index.
Discussions:
7. The discussion needs to be improved. I would suggest authors add more discussions about the effects of HIIT on affect, attention, and physiology, and the association between these factors.
8. The present article is lacking limitations and future directions.
Conclusions:
9. The conclusions need to be improved.
10. According to the results, authors concluded that affective responses during this type of HIIT are independent of aerobic fitness. In my opinion, authors could make conclusions in another view, such as “participants could benefit from the acute Tabata HIIT regardless they are high fitness even low fitness”, that means the acute Tabata HIIT is the important and effective, and this exercise mode could promote to benefit more people.
References:
11. Reference list for such kind of scientific article is too short, with just 44 references. And the latest literature is too few, accounting for only 1/4 in the last 5 years.
12. Authors need to add doi to all of the references.
13. Reference [13] may not support the statement “An important determinant of the physiological and metabolic responses to a HIIT session may be aerobic fitness [13, 14].
Minor concerns:
Methods:
- I missed the detailed informations about the blood lactate concentration.
- Authors need to report the reliability and validity of each questionnaires.
- Supported references that why the authors used the rhythm and sound of the music in the experiment are needed.
- Please add the calculation formula of aerobic fitness (page 3, 2.3), the values of the percentile norms set by ACSM (page 4, 2.3) as well as the calculation formula and references of the peak heart rate (page 4, 2.3). In addition, please check all of the “HRpeak” in the full text, some of them were presented by Hrpeak.
- Please add more information (brand, manufacture) about the HR monitor.
- Please add the statistical significance set in 2.4.
- Spelling mistake: (iii) two-aay.
Results:
- Please report all of the main effects and the interaction effects, even if there is no significant difference (3.2 and 3.3).
- ***p < 0.001 was not consistent with **p in Figure 1 and **p < 0.001 in Figure 4. Additionally, all of the p values should be italic and nonbold (please check the full text).
- Please add η2p (effect size) in the results.
- Please report whether the authors used standard deviation or standard error in Figures by notes.
- The positions of * could easily mislead readers.
- The descriptions of Figure 1, 2, 3 and 5 were too simple, please add more details.
- Please check the values of F(xx,xx) (page 5, 3.1; page 6, 3.3).
- Please add the *or **or*** in Table 2 and notes if there is a significant difference.
- “HF”, “pre-test” and “M±SD” in Table 2 should not be bold.
Author Response
Reviewer 1
Thank you very much for your kind feedback and important notes and suggestions that helped us improve our study.
Major concerns:
Introductions:
- Authors need to add more details and references in this part, such as the literatures about the effects of acute HIIT or acute Tabata HIIT on affective, cognitive and physiological indicators. Besides, the research gap needs to add in the last paragraph, before the purpose of this study.
Thank you for pointing out this gap in the literature review of our Introduction. More details were added in all the 3 paragraphs of the Introduction based on your comment.
The following sentences were added in page 2/13, paragraph 1 of Introduction, lines* 7-12 “Tabata consists of 20 s of high intensity effort followed by 10 s of passive recovery repeated for eight rounds. This training protocol has appeared to be safe and effective even in sedentary populations, improving blood biomarkers, increasing caloric expenditure and promoting both aerobic and anaerobic fitness [3-5]. Nevertheless, conflicting data has supported the notion that Tabata should be mainly addressed to athletes [6].
The following sentences were added in page 2/13, paragraph 2 of Introduction, lines 7-10 “The increased concentration of blood lactate combined with the increased oxygen consumption is metabolically unsustainable and causes unpleasant feelings to the participants urging them to stop the exercise [39].”
The following sentences were added in page 2/13, paragraph 3 of Introduction, lines 4-7 “Moreover, HIIT accelerates and enhances the metabolic benefits of exercise and, secondly, it reduces the time required to commit to training [40]. However, strenuous and excessive exercise may adversely affect cognitive function [48].”
- Why did the authors choose “healthy young women” as the participant? Please explain and add supported references.
A more thorough explanation for the reason why we only recruited healthy young women was added starting in the 2nd sentence of Discussion and on.
The following sentences were added in page 9/13, paragraph 1 of Discussion, lines 3-7 “Sex-specific effects of training performance during HIIT protocols have not been adequately investigated, despite the anthropometric and physiological differences between males and females. Moreover, some studies have investigated female samples without controlling the menstrual cycle, thus impairing the generalization of data for women [33, 37, 38].”
Methods:
- Why did the experiment design lack a control group?
The present study did not involve an intervention, but aimed to compare physiological, psychological and cognitive responses to a high intensity interval workout (Tabata) between women with higher and low aerobic fitness. Therefore, a control group was not necessary for this design.
- Why the authors used a lot of affect-related questionnaires? Which questionnaire could reflect the affect most?
Thank you for your comment. We have excluded the Positive Affect – Negative Affect Schedule from the paper, as ADACL and POMS are the most reporesentative instruments the affective responses of the participants, and there is no need to include the PANAS, and FAS.
- Why the authors examined the visual focus? Does this have an association with affect? Authors should add reasons in the introduction part or in the methods part. Besides, could CGT task represent cognitive function? Maybe regard this as attention is more appropriate. Authors could also consider revising the title.
Thank you and we agree for your comment. We have changed the title, and also added the description of the test based on your comment.
Results:
- I would suggest authors examine the correlations among affect, attention, and physiological index.
Based on your comment we have added a section in the results.
3.7. Correlation among physiological indices, attention, and affective responses
The relationship among physiological indices (HRpeak, blood lactate concentration), and attention (GRD), with participants’ affective responses were also examined. The results didn’t show any significant correlation, except between RPE and fatigue (r = .53, p<.01), and RPE and vigor (r = -.46, p<.05). In the post-test measure, blood lactate concentration showed positive correlation with calmness (r = .42, p<.05), and negative correlation with tension (r = -.42, p<.05). CGT showed negative correlation to fatigue (r = -39, p<.05). Finally, RPE showed negative correlations to vigor (POMS), energy (ADACL) (r = -.51, p<.01, r = -.62, p<.001), and positive correlations to anger(r = .44, p<.05), fatigue (r = .67, p<.001), confusion (r = .47, p<.05), and TMD (r = .72, p<.001).
Discussions:
- The discussion needs to be improved. I would suggest authors add more discussions about the effects of HIIT on affect, attention, and physiology, and the association between these factors.
Discussion has been updated. The following sentences were added in page 9/13, paragraph 1 of Discussion, lines 3-10 “Sex-specific effects of training performance during HIIT protocols have not been ade-quately investigated, despite the anthropometric and physiological differences between males and females. Moreover, some studies have investigated female samples without controlling the menstrual cycle, thus impairing the generalization of data for women [33, 37, 38]. Although any woman could voluntarily participate in the study, the sample was not totally randomized. In that way the external validity of our study is limited. Therefore, future designs are encouraged to randomly recruit both men and women and match their physical fitness parameters.” However, the structure of our Discussion is in line with your suggestion since first of all, we explain our findings regarding the physiological factors, secondly, we elaborate on the results about affect and emotions and in the end, we discuss our findings about concentration. Throughout those paragraphs, the interaction and association among those factors has been discussed and explained in relation to existing literature.
- The present article is lacking limitations and future directions.
Limitations and future directions were added in Discussion starting from the 2nd sentence and on. The following sentences were added in page 9/13, paragraph 1 of Discussion, lines 3-10 “Sex-specific effects of training performance during HIIT protocols have not been ade-quately investigated, despite the anthropometric and physiological differences between males and females. Moreover, some studies have investigated female samples without controlling the menstrual cycle, thus impairing the generalization of data for women [33, 37, 38]. Although any woman could voluntarily participate in the study, the sample was not totally randomized. In that way the external validity of our study is limited. Therefore, future designs are encouraged to randomly recruit both men and women and match their physical fitness parameters.” Moreover, in the last sentence in “5. Conclusions” another suggestion for future studies had already been stated in the first version of our article already “Future studies should examine the psychological effects of a long-term (over eight weeks) HIIT protocol executed by both men and women of different physical fitness level, which should be assessed using more indices, such as the respiratory or lactate threshold”.
Conclusions:
- The conclusions need to be improved.
Conclusions have been updated! The following sentences were added in pages 10-11/13, paragraph 1 of Conclusions, lines 7-12 “The most important finding of this study, which promotes the knowledge concerning HIIT, is that the sample was split into two experimental groups based on their level of physical fitness and while the two groups showed differences in VO2max and possibly in the associated physiological parameters -respiratory and lactate thresholds-, their psy-chological responses did not show robust differences as expected.”
- According to the results, authors concluded that affective responses during this type of HIIT are independent of aerobic fitness. In my opinion, authors could make conclusions in another view, such as “participants could benefit from the acute Tabata HIIT regardless they are high fitness even low fitness”, that means the acute Tabata HIIT is the important and effective, and this exercise mode could promote to benefit more people.
This is a very good point since indeed our conclusions lacked of the optimistic point of view. The update in this section is based on your comment, since we have now tried to underline the most important and novel finding. The following sentences were added in pages 10-11/13, paragraph 1 of Conclusions, lines 7-12 “The most important finding of this study, which promotes the knowledge concerning HIIT, is that the sample was split into two experimental groups based on their level of physical fitness and while the two groups showed differences in VO2max and possibly in the associated physiological parameters -respiratory and lactate thresholds-, their psy-chological responses did not show robust differences as expected.”
References:
- Reference list for such kind of scientific article is too short, with just 44 references. And the latest literature is too few, accounting for only 1/4 in the last 5 years.
6 more articles were added to the reference list
More specifically References [3-6] and [37, 38]
- Authors need to add doi to all of the references.
All references now are updated and have a doi!
- Reference [13] may not support the statement “An important determinant of the physiological and metabolic responses to a HIIT session may be aerobic fitness [13, 14].
-Reference number has been updated since more references were added.
[17] Bartlett, J.D., Close, G.L., MacLaren, D.P., Gregson, W., Drust, B., & Morton, J.P. High-intensity interval running is perceived to be more enjoyable than moderate-intensity continuous exercise: implications for exercise adherence. Journal of Sports Sciences 2011, 29, 547-553. https://doi.org/10.1080/02640414.2010.545427
In this study, the authors have suggested that high-intensity exercise may be more enjoyable than moderate-intensity exercise but maybe this conclusion has been conducted due to the fact that the sample consisted of fitter individuals with better aerobic fitness. As so, aerobic fitness seems to be an important determinant of the physiological responses to a HIIT session.
Minor concerns:
Methods:
- I missed the detailed informations about the blood lactate concentration.
Blood lactate concentration information can be found in page 6/13, section 3.2., paragraph 1, lines 1-5 “The analysis of variance showed a statistically significant main effect of time [F (2, 52) = 195.859, p < .001]. Pairwise comparisons revealed significant difference between pre-test to mid-test (p < .001) as well as between pre-test to post-test measures (p < .001). Blood lactate concentration peaked at mid-test and remained at the same level at post-test measure (Figure 2).” and is explained furtherly in the last 11 lines of the 1st paragraph of Discussion. The following sentences were added in page 9/13, paragraph 1 of Discussion, lines 17-21 “Furthermore, blood lactate concentration was increased to around 12 mM after the third round of the protocol and remained at that level until the end of exercise in both groups (Fig. 2). The similar HR and blood lactate in the two groups, irrespective of different aerobic fitness has also been recently reported during HIIT, where no correlation was found between VO2max levels and blood lactate concentration [39].”
- Authors need to report the reliability and validity of each questionnaires.
The reliability and validity indices have been added, as well as from previous studies.
- Supported references that why the authors used the rhythm and sound of the music in the experiment are needed.
The Tabata workout format has its origins in the paper of Tabata et al. (1996), where participants trained for 6 weeks, five days per week, with a cycling protocol involving 7-8 sets of 20-s exercise at an intensity of about 170% of VO2max with a 10-s rest between each bout. This training increased both aerobic and anaerobic power (VO2max increased by 7 ml.kg-1.min-1, while the anaerobic capacity increased by 28%). This workout format has been modified by physical trainers in terms of intensity (it was high, but not maximal) and exercise modality (various exercises are used instead of cycling, such as burpees, lunges etc.). Nowdays, the only characteristic kept is the high-intensity character and the workout “format”, i.e., a Tabata workout is performed for 20-30 min, with each “round” consisting of eight sets of fast-paced exercises performed for 20 s, interspersed with a brief rest of 10 s (https://www.merriam-webster.com/words-at-play/what-does-tabata-mean-hiit ).To make this type of training suitable for exercise in the gym, music is added, and the rhythm varies from 96 beats per minute (i.e., 16 repetitions of the exercise per 20 s), to 144 beats per minute (i.e., 24 repetitions of the exercise per 20 s) (https://open.spotify.com/track/48ExFvRdKWkQ3Thxm1rV8H). To date there is no study examining the effects of rhythm on physiological or psychological responses to the Tabata workout, but faster execution increases the physiological demands of the workout and is suitable for more advanced individuals
- Please add the calculation formula of aerobic fitness (page 3, 2.3), the values of the percentile norms set by ACSM (page 4, 2.3) as well as the calculation formula and references of the peak heart rate (page 4, 2.3).
In addition, please check all of the “HRpeak” in the full text, some of them were presented by Hrpeak.
Corrected it!
- Please add more information (brand, manufacture) about the HR monitor.
Corrected it! The requested information was added in page 4/13, paragraph 4 of Procedures, line 6 “(Polar Electro Oy, Kempele, Finland)”.
- Please add the statistical significance set in 2.4.
Ok, it has been added in 2.4. section.
- Spelling mistake: (iii) two-aay.
Corrected it!
Results:
- Please report all of the main effects and the interaction effects, even if there is no significant difference (3.2 and 3.3).
It has been added in the text, where is was messing.
- ***p < 0.001 was not consistent with **p in Figure 1 and **p < 0.001 in Figure 4. Additionally, all of the p values should be italic and nonbold (please check the full text).
Thanks for the comment. It has been changed and corrected in the text.
- Please add η2p (effect size) in the results.
It has been added in the text.
- Please report whether the authors used standard deviation or standard error in Figures by notes.
It has been added in the figures so as to be clear for the reader.
- The positions of * could easily mislead readers.
Ok corrected.
- The descriptions of Figure 1, 2, 3 and 5 were too simple, please add more details.
It has been added more details in the Figures 1, 2, 3, and 5.
- Please check the values of F(xx,xx) (page 5, 3.1; page 6, 3.3).
Thanks fo the comment. It has changed.
- “HF”, “pre-test” and “M±SD” in Table 2 should not be bold.
Corrected it!

Reviewer 2 Report
The Effect of Aerobic Fitness on Psychological, Cognitive and Physiological Responses during a Tabata High Intensity Interval Training Session in Healthy Young Women
First of all, the reviewer would like to thank the authors for their work and efforts in trying to improve sports science knowledge.
General comments to the authors
Overall, this is a nice study that could have great practical application. The authors are commended on their efforts thus far. The study is well designed and well-written, with a great original article evaluating the usefulness of the topic. However, I suggest only small corrections for manuscript.
Abstract
This section is well designed and well-written.
Introduction section
This section is short but it is enough.
Methods section
The authors should add effect sizes with descriptions in Statistical Analysis results and tables.
Results section
This section is well designed and well-shown.
Discussion section
Overall the discussion is well-written and incorporates relevant literature.
The authors should add these important articles about tabata, hiit and psychophysiological responses to support their ideas
Ramírez-Marrero, F. A., Trinidad, J., Pollock, J., Casul, Á., & Bayrón, F. E. (2014). Testing Tabata High-Intensity Interval Training Protocol in Hispanic Obese Women. Journal of Women’s Health Physical Therapy, 38(3), 99-103.
Olson, M. (2014). TABATA: It’sa HIIT!. ACSM'S Health & Fitness Journal, 18(5), 17-24.
Emberts, T., Porcari, J., Dobers-Tein, S., Steffen, J., & Foster, C. (2013). Exercise intensity and energy expenditure of a tabata workout. Journal of sports science & medicine, 12(3), 612.
Arslan, E., Can, S., & Demirkan, E. (2017). Effect of short-term aerobic and combined training program on body composition, lipids profile and psychological health in premenopausal women. Science & Sports, 32(2), 106-113.
Olson, M. (2013). Tabata interval exercise: energy expenditure and post-exercise responses. Med Sci Sports Exerc, 45, S420.
Author Response
Reviewer 2
Thank you very much for your kind feedback and important notes and suggestions that helped us improve our study.
Abstract
This section is well designed and well-written.
Thank you!
Introduction section
This section is short but it is enough.
Thank you!
Methods section
The authors should add effect sizes with descriptions in Statistical Analysis results and tables.
Thank you for the comment. The Effect syzes have been added in the statistical analysis.
Results section
This section is well designed and well-shown.
Thank you!
Discussion section
Overall the discussion is well-written and incorporates relevant literature.
Thank you!
The authors should add these important articles about tabata, hiit and psychophysiological responses to support their ideas
Ramírez-Marrero, F. A., Trinidad, J., Pollock, J., Casul, Á., & Bayrón, F. E. (2014). Testing Tabata High-Intensity Interval Training Protocol in Hispanic Obese Women. Journal of Women’s Health Physical Therapy, 38(3), 99-103.
Olson, M. (2014). TABATA: It’sa HIIT!. ACSM'S Health & Fitness Journal, 18(5), 17-24.
Emberts, T., Porcari, J., Dobers-Tein, S., Steffen, J., & Foster, C. (2013). Exercise intensity and energy expenditure of a tabata workout. Journal of sports science & medicine, 12(3), 612.
Arslan, E., Can, S., & Demirkan, E. (2017). Effect of short-term aerobic and combined training program on body composition, lipids profile and psychological health in premenopausal women. Science & Sports, 32(2), 106-113.
Olson, M. (2013). Tabata interval exercise: energy expenditure and post-exercise responses. Med Sci Sports Exerc, 45, S420.
Thank you for pointing out those interesting articles. Useful information was collected and added in the last sentences of the 1stparagraph of Introduction from 4/5 articles mentioned (the ones underlined above). The following sentences were added in page 2/13, paragraph 1, lines 7-12 “Tabata consists of 20 s of high intensity effort followed by 10 s of passive recovery repeated for eight rounds. This training protocol has appeared to be safe and effective even in sedentary populations, improving blood biomarkers, increasing caloric expenditure and promoting both aerobic and anaerobic fitness [3-5]. Nevertheless, conflicting data has supported the notion that Tabata should be mainly addressed to athletes [6].” derived from the articles you kindly suggested.
Reviewer 3 Report
Thank you for this very interesting study. I will begin by summarizing the study in my own words, and then provide specific comments below to allow for easier understanding. This is a cross-sectional study in which female participants underwent a high-intensity training program and responded to questionnaires regarding psychological and cognitive factors, as well as physiological measures, prior to, during, and after training. Results indicate that those with high fitness and low fitness are not statistically different in many of these measures, and the authors conclude that aerobic fitness level does not influence affective responses to training.
1. I believe a slightly more thorough explanation of the relevance of the Dual-Mode Theory (i.e. potential mechanisms or more practical consequences) might aid the reader in understanding the aim of this study.
2. It may be relevant to provide an explanation of “Tabata” to clarify the connection to HIIT.
3. I believe it would be relevant for the authors to explain why only women were included in the sample.
4. In the last sentence in the POMS-SF paragraph: unnecessary comma following “as well as”
5. In the first sentence in the PANAS paragraph: either “comprising 20 items” or “comprised of 20…”
6. In the POMS-SF and RPE paragraphs: The authors may consider removing the period after the name and combine the first and second sentences. This may help with readability.
7. The purpose of having two testing occasions is slightly unclear. Is the first occasion merely to test aerobic capacity in order to categorize as high or low fitness levels, or were there other measurements relevant to the study’s aim? I believe this could be made clearer.
8. While performing the tabata, were the three (or four) questionnaires (and lactate test) able to be completed within the 1 (or two) minute rest period? There may be an argument for the idea that the cognitive load of filling out questionnaires may interfere with the rest.
a. Along the same lines, what is meant by a “passive rehabilitation break?”
9. When explaining the exercises and their order: “were performed in that order” is grammatically odd. I would recommend writing “in the following order” or something similar.
10. Was maintenance of 70% peak heart rate monitored by the instructor or was it self-monitored by the participant? Was there an acceptable range defined?
11. Statistical analysis: “(iii) a two-aay ANOVA” should be corrected to “way.”
12. In the two-way ANOVA (2 Fitness Levels x 8 Time points), I am slightly confused as to the 8th time-point, as it would appear from the description of the method and Figure 1 that FS, FAS, and RPE are only measured 7 times.
a. It appears from the results section that a measurement was taken following Tabata No. 6, yet prior to the post-test. Is this correct? How long after the final tabata/rest was the post-test completed?
13. Results 3.5 Felt Arousal Scale – Is it relevant to report that the results show a “marginally statistically significant” difference? I understand there is a difference which is close to significance, but this may be seen as problematic.
14. Results 3.8 Activation-Deactivation Adjective Check List: “calmness factors, the two-way ANOVA 2x2 for showed” – I would recommend deleting “for” in this sentence.
15. Discussion: MICT is written without an explanation of the acronym.
16. I believe it would be relevant to discuss strengths and limitations in the discussion section. For example, in relation to whether the exercise is self-paced or prescribed. Ekkekakis (Ekkekakis P. Let them Roam Free? Sports Medicine. 2009; 39(10):857-88), for example, indicates that this may have an effect on perceived exertion, which may in turn reflect on other psychological and cognitive aspects of physical activity. There may also be aspects to discuss in terms of the generalizability to a larger population given the limited sample tested in this study.
17. Overall discussion: I find this to be a very interesting discussion, and one which I believe deserves a lot of attention in the literature. I wonder if the authors considered the effect of physical activity on cognitive function based on things like enjoyment or motivation. For example, Diekfuss (Diekfuss et al. Targeted Application of Motor Learning Theory to Leverage Youth Neuroplasticity for Enhanced Injury-Resistance and Exercise Performance: OPTIMAL PREP. Journal of Science and Sport in Exercise. 2021;3(1):17-36) discuss the dopaminergic system, among other things, as potential factors which may influence the way in which exercises are performed. I believe motivation as a whole may also be interesting to discuss, as a person not motivated to train may have less positive reactions.
18. Conclusion: There is a clear conclusion, but there is also a degree of repetition of results. I would recommend clarifying the main take-away message of this study.
Author Response
Reviewer 3
Thank you very much for your kind feedback and important notes and suggestions that helped us improve our study.
- I believe a slightly more thorough explanation of the relevance of the Dual-Mode Theory (i.e. potential mechanisms or more practical consequences) might aid the reader in understanding the aim of this study.
Good point! We added it in the 2nd paragraph of Introduction. The following sentences were added in page 2/13, paragraph 2, lines 7-10 “The increased concentration of blood lactate combined with the increased oxygen con-sumption is metabolically unsustainable and causes unpleasant feelings to the partici-pants urging them to stop the exercise [39].”
- It may be relevant to provide an explanation of “Tabata” to clarify the connection to HIIT.
A further explanation was added in the last sentences of the 1st paragraph of Introduction. The following sentences were added in page 2/13, paragraph 1, lines 7-12 “Tabata consists of 20 s of high intensity effort followed by 10 s of passive recovery repeated for eight rounds. This training protocol has appeared to be safe and effective even in sedentary populations, improving blood biomarkers, increasing caloric expenditure and promoting both aerobic and anaerobic fitness [3-5]. Nevertheless, conflicting data has supported the notion that Tabata should be mainly addressed to athletes [6].”
- I believe it would be relevant for the authors to explain why only women were included in the sample.
The explanation of this limitation of the sample has been added in the 2nd sentence of the Discussion. The following sentences were added in page 9/13, paragraph 1, lines 3-7 “Sex-specific effects of training performance during HIIT protocols have not been ade-quately investigated, despite the anthropometric and physiological differences between males and females. Moreover, some studies have investigated female samples without controlling the menstrual cycle, thus impairing the generalization of data for women [33, 37, 38].”
- In the last sentence in the POMS-SF paragraph: unnecessary comma following “as well as”
Corrected it.
- In the first sentence in the PANAS paragraph: either “comprising 20 items” or “comprised of 20…”
Corrected it.
- In the POMS-SF and RPE paragraphs: The authors may consider removing the period after the name and combine the first and second sentences. This may help with readability.
Corrected it.
- The purpose of having two testing occasions is slightly unclear. Is the first occasion merely to test aerobic capacity in order to categorize as high or low fitness levels, or were there other measurements relevant to the study’s aim? I believe this could be made clearer.
The procedures that were conducted in the 1st occasion are explained in the 2nd sentence of 2.3. “During the first visit, the participants filled in a demographic information form and they were asked about their previous experience in HIIT programs, and their height and weight measured. Afterwards, they performed the 20-m Shuttle Run Test in an indoor hall, after test familiarization.”. The reason why we had to add a first visit was to collect the demographic data and to get their height and weight measured. Mainly, though, the 1st visit was necessary in order to perform the aerobic fitness test on a different day than the main measurements, because this test lead to the categorization of the sample groups, which was necessary as indicated by the main objective of the study – to compare high vs low fitness participants.
- While performing the tabata, were the three (or four) questionnaires (and lactate test) able to be completed within the 1 (or two) minute rest period? There may be an argument for the idea that the cognitive load of filling out questionnaires may interfere with the rest.
The 3 scales that were administered in the 1-min/2-min rest are single-item, which means that the participants replied to them using one word (specifically one number!) and took them approximately 5 s to do so. A clarification was added in the manuscript. The following clarification was added in page 5/13, paragraph 4 of Procedures, line 27 “in one word”. The rest in mainly important for the partial physical rehabilitation of the participants.
- Along the same lines, what is meant by a “passive rehabilitation break?”
Passive rehabilitation break refers to the absence of physical activity throughout the resting period. The clarification was added in page 5/13, paragraph 4 of Procedures, line 14 “(no physical activity)”. Opposite (active rehabilitation) would be to add a low intensity exercise during the break.
- When explaining the exercises and their order: “were performed in that order” is grammatically odd. I would recommend writing “in the following order” or something similar.
Corrected it.
- Was maintenance of 70% peak heart rate monitored by the instructor or was it self-monitored by the participant? Was there an acceptable range defined?
A clarification was added in that point. The following restatement was added in page 4/13, paragraph 4 of Procedures, line 11 “guiding them, also, to keep”. It was monitored by the experienced instructor who gave them guidelines during the warm-up. This percentage is approximate as it only refers to the warm-up and not the main protocol and we tried to meet that criterion since it was suggested and used in reference No. 30.
- Statistical analysis: “(iii) a two-aay ANOVA” should be corrected to “way.”
Corrected it.
- In the two-way ANOVA (2 Fitness Levels x 8 Time points), I am slightly confused as to the 8thtime-point, as it would appear from the description of the method and Figure 1 that FS, FAS, and RPE are only measured 7 times.
- It appears from the results section that a measurement was taken following Tabata No. 6, yet prior to the post-test. Is this correct? How long after the final tabata/rest was the post-test completed?
The following clarification was added in page 4/13, paragraph 4 of Procedures, line 30-31 “Upon completion of the training protocol and after the single-item questionnaires were directly answered again (FS, FAS, RPE), each participant completed again, approximately 1-min after the completion of the training protocol”. The participants replied to the single-item scales after the Tabata No. 6 (like they did after all the previous Tabata) -Time Point 7- and approximately after 1-min they completed all of the post measurements -Time Point 8-, which included again those single-item scales.
- Results 3.5 Felt Arousal Scale – Is it relevant to report that the results show a “marginally statistically significant” difference? I understand there is a difference which is close to significance, but this may be seen as problematic.
Thank you for your comment. However, the Felt Arousal Scale has been excluded from the analysis based on the comments of reviewer 1, as there were enough instrument evaluating the affective responses of the participants.
- Results 3.8 Activation-Deactivation Adjective Check List: “calmness factors, the two-way ANOVA 2x2 for showed” – I would recommend deleting “for” in this sentence.
Corrected it.
- Discussion: MICT is written without an explanation of the acronym.
Corrected it. The following clarification was added in page 10/13, paragraph 2 of Discussion, line 3-4 “moderate-intensity continuous training”.
- I believe it would be relevant to discuss strengths and limitations in the discussion section. For example, in relation to whether the exercise is self-paced or prescribed. Ekkekakis (Ekkekakis P. Let them Roam Free? Sports Medicine. 2009; 39(10):857-88), for example, indicates that this may have an effect on perceived exertion, which may in turn reflect on other psychological and cognitive aspects of physical activity. There may also be aspects to discuss in terms of the generalizability to a larger population given the limited sample tested in this study.
In the 2nd sentence in the Discussion section text was added referring to this comment. The following sentences were added in page 9/13, paragraph 1, lines 3-10 “Sex-specific effects of training performance during HIIT protocols have not been ade-quately investigated, despite the anthropometric and physiological differences between males and females. Moreover, some studies have investigated female samples without controlling the menstrual cycle, thus impairing the generalization of data for women [33, 37, 38]. Although any woman could voluntarily participate in the study, the sample was not totally randomized. In that way the external validity of our study is limited. Therefore, future designs are encouraged to randomly recruit both men and women and match their physical fitness parameters.“. Thank you for indicating it!
- Overall discussion: I find this to be a very interesting discussion, and one which I believe deserves a lot of attention in the literature. I wonder if the authors considered the effect of physical activity on cognitive function based on things like enjoyment or motivation. For example, Diekfuss (Diekfuss et al. Targeted Application of Motor Learning Theory to Leverage Youth Neuroplasticity for Enhanced Injury-Resistance and Exercise Performance: OPTIMAL PREP. Journal of Science and Sport in Exercise. 2021;3(1):17-36) discuss the dopaminergic system, among other things, as potential factors which may influence the way in which exercises are performed. I believe motivation as a whole may also be interesting to discuss, as a person not motivated to train may have less positive reactions.
Definitely enjoyment is a variable worth examining as well as motivation but since we used many scales and questionnaires in this study, we tried not to add more, even though more were under consideration when designing the study, because then it would be hard to have a sufficient sample size without statistical power problems.
- Conclusion: There is a clear conclusion, but there is also a degree of repetition of results. I would recommend clarifying the main take-away message of this study.
In the Conclusions section, information was added, which sums up the most important outcome of this study. The following sentences were added in page 10-11/13, paragraph 1 on Conclusion, lines 7-12 “The most important finding of this study, which promotes the knowledge concerning HIIT, is that the sample was split into two experimental groups based on their level of physical fitness and while the two groups showed differences in VO2max and possibly in the associated physiological parameters -respiratory and lactate thresholds-, their psy-chological responses did not show robust differences as expected.”